# *In Vitro* Antioxidant Activity of Areca Nut Polyphenol Extracts on RAW264.7 Cells

**DOI:** 10.3390/foods11223607

**Published:** 2022-11-12

**Authors:** Shuhan Yi, Luyan Zou, Zongjun Li, Kozue Sakao, Yuanliang Wang, De-Xing Hou

**Affiliations:** 1The United Graduate School of Agricultural Sciences, Kagoshima University, Kagoshima 890-0065, Japan; 2College of Food Science and Technology, Hunan Agricultural University, Changsha 410128, China; 3Hunan Province Key Laboratory of Food Science and Biotechnology, Changsha 410128, China; 4National Engineering Center of Plant Functional Components Utilization, Changsha 410128, China; 5Faculty of Agriculture, Kagoshima University, Kagoshima 890-0065, Japan

**Keywords:** areca nut polyphenol, ROS, Nrf2, HO-1, RNA-seq

## Abstract

Chewing areca nuts is a popular hobby in the Asian region, and areca nuts are rich in polyphenols, although some alkaloids are included. In this study, we explored the antioxidant activity of areca nut polyphenols (ANP) in lipopolysaccharides (LPS)-stimulated RAW264.7 cells. The results revealed that ANP reduced the level of reactive oxygen species (ROS) in LPS-stimulated RAW264.7 cells and enhanced the expression of nuclear factor erythroid 2-related factor 2 (Nrf2) and heme oxygenase 1 (HO-1). RNA-seq analysis showed that ANP down-regulated the transcription of genes related to the cancer pathway at 160 μg/mL, and the inflammatory pathway as well as viral infection pathway at 320 μg/mL. The cellular signaling analysis further revealed that the expressions of these genes were regulated by the mitogen-activated protein kinase (MAPK) pathway, and ANP downregulated the activation of the MAPK signaling pathway stimulated by LPS. Collectively, our findings showed that ANP inhibited the MAPK pathway and activated the Nrf2/HO-1 antioxidant pathways to reduce ROS generation induced by LPS.

## 1. Introduction

*Areca catechu* L. is a tropical fruit widely grown in China, India, and Southeast Asia [1]. Historically, it has been used as a chewing substance, and it possesses a certain addictive nature. Moreover, the areca nut has been used in traditional Chinese medicine in China and Ayurvedic medicine in India, since the areca nut can remove parasites from the body and disperse effusions in the abdominal cavity [2,3]. On the other hand, the International Agency for Research on Cancer classified areca nut as a Class I carcinogen in 2003 because it contains a large number of alkaloids, such as arecoline, arecaidine, and guvacine, and these arecolines are carcinogenic [4,5]. The safety of areca nuts has become a hotly debated topic [6]. Therefore, the new direction of areca nut product applications is to separate the alkaloids from the polyphenols and further focus on the research and development of the polyphenols in the areca nut.

The antioxidant effects of plant polyphenols have received a lot of interest in recent years, since free radical reactive oxygen species (ROS) are often produced when the body is exposed to different biochemical conditions or pathological states. Therefore, the ability to remove ROS is an important part of antioxidants. Several studies have shown that areca leaf and areca nut extracts have antioxidant and radical scavenging activities [7,8]. However, the components of the areca nut contributing those antioxidant effects are unclear. Areca nut contains two kinds of major bioactive compounds, which are polyphenols and alkaloids. Alkaloids have been reported to be carcinogenic. RAW264.7 is a macrophage-like cell line, which is often used as a cellular model to study antioxidant and anti-inflammatory activities at the cellular level. Especially, the stimulation of lipopolysaccharides (LPS) can lead to the production of ROS and inflammation in RAW264.7 cells [9,10]. Therefore, in this study, we used RAW264.7 cells to investigate how areca nut polyphenols play a role against LPS-induced ROS and inflammation at the cellular level.

The balance between the generation and elimination of cellular ROS is critically regulated by cellular antioxidant enzyme genes containing the antioxidant/electrophile-responsive element (ARE/EpRE) in their promoters. Several molecules, such as nuclear factor erythroid 2-related factor 2 (Nrf2), transcription factor Jun (c-Jun), activating transcription factor 2 (ATF2), and activating transcription factor 4 (ATF4), are possible modulators of ARE/EpRE [11]. Of these, Nrf2 is a powerful activator of ARE-mediated gene expression that regulates the expression of several antioxidant enzyme genes, including heme oxygenase 1 (HO-1) [12]. Nrf2 could be activated by both Keap1-dependent and Keap1-independent pathways. The phosphorylation of protein kinases C, MAPKs, and PI3K is closely related to Nrf2 activity in a Keap1-independent way [13,14,15,16,17,18].

High-throughput RNA-sequencing technology has been shown to be an effective tool for studying the bioactive effects and mechanisms of natural compounds at the genome-wide gene expression level [19,20,21]. Thus, we ultimately used this technology to evaluate the molecular mechanisms of antioxidant activity of APN in LPS-stimulated RAW264 cells.

## 2. Materials and Methods

### 2.1. Materials

LPS were purchased from Sigma (St. Louis, MO, USA). Dulbecco’s modified eagle media (DMEM) and fetal bovine serum (FBS) were purchased from Biological Industries (Kibbutz Beit Haemek, Israel). MTT cell proliferation and cytotoxicity assay kits were purchased from Solarbio (Beijing, China). Reactive oxygen species assay kits were purchased from Beyotime (Shanghai, China). Anti-Nrf2 antibody (Nrf2), anti-Heme Oxygenase 1 antibody (HO-1), and anti-GAPDH antibody (GAPDH) were purchased from Abcam (Cambridge, UK); anti-MEK1 antibody (MEK1), anti-p-MEK1 antibody (p-MEK1), anti-ERK antibody (ERK), anti-p-ERK antibody (p-ERK), anti-p-MKK3 antibody (p-MKK3), anti-p38 antibody (p38), anti-p-p38 antibody (p-p38), anti-MKK4 antibody (MKK4), anti-p-MKK4 antibody (p-MKK4), anti-JNK antibody (JNK), anti-p-JNK antibody (p-JNK), anti-α-tubulin antibody (α-tubulin), and anti-rabbit antibody were purchased from CST (Boston, MA, USA).

### 2.2. Preparation of Areca Nut Polyphenols

The raw materials of the areca nut, which was named by James A. Duke (Agricultural Research Service, USDA), were obtained in January 2018 from Wanning city, Hainan, China. Areca nut polyphenols (ANP) were extracted as described previously. In brief, fresh areca nut was freeze-dried and ground into powder. The crude polyphenol from areca nut powder was extracted with 50% (*v*/*v*) ethanol at 68 °C for 48 min and then freeze-dried. The crude polyphenols were purified with XAD-7 macroporous resin by washing them with 50% ethanol. The total polyphenol contents were estimated at 80% by the Folin-Ciocalteu method [22]. In previous studies, the two kinds of polyphenol compositions of areca nuts were catechins (2060.44 ± 18.24 μg/mL) and proanthocyanidin B1 (2510.18 ± 62.40 μg/mL) [23].

### 2.3. Antioxidant Activity Assays in LPS-Stimulated RAW264.7 Macrophages

#### 2.3.1. Cell Culture

RAW264.7 macrophages were obtained from the American Type Culture Collection (VA, USA) and cultured in DMEM containing 10% FBS and 1% antibiotic at 37 °C in 5% CO_2_ humidified air.

#### 2.3.2. Cell Viability Assay

Cell viability was assessed using the MTT cell proliferation and cytotoxicity assay kit [24]. Briefly, RAW 264.7 cells (2 × 10^4^ cells/well) were incubated into each well of a 96-well plate. After 24 h of culture, they were treated with different concentrations (40, 80, 160, 320, and 640 μg/mL) of ANP at different times (6, 9, 12, and 15 h). Then the culture medium was removed, and 90 μL of the new medium and 10 μL of MTT solution were added to continue the culture for 4 h. After the culture medium was removed, 150 μL culture medium was added and shaken at low speed for 10 min on a shaker with light protection and detected at 490 nm on a microplate reader (Thermo Scientific Multiskan, version 1.00.79, Vantaa, Finland). The absorbance of the ANP-treated group was divided by the absorbance of the control group to determine the percentage of cell viability.

#### 2.3.3. Cellular ROS Detection by 2′,7′-Dichlorodihydrofluorescein Diacetate and Fluorescence Inverted Microscope Observation

The intracellular formation of ROS was detected using the fluorescent probe 2’,7’-dichlorodihydrofluorescein diacetate (DCFH-DA). This compound easily passes the cellular membrane into cells and is hydrolyzed by intracellular esterase to yield 2’,7’-dichlorodihydrofluorescein (DCFH). ROS produced by the cells oxidizes DCFH to the highly fluorescent compound 2’,7’-dichlorofluorescein (DCF). Thus, the fluorescence intensity is proportional to the amount of ROS produced by the cells. RAW264.7 cells were plated in 96-well plates pre-incubation for 24 h; the cells were then stimulated by LPS for 30 min, then treated with or without ANP for 12 h. After the prepared DCFH-DA was added to the 96-well plates at 10 μM and incubated for 30 min at 37 °C. Then the cells were washed three times with serum-free media. The fluorescence intensity was measured at an excitation wavelength of 488 nm and an emission wavelength of 530 nm using a fluorescence counter (FL 6500, Perkin-Elmer, Waltham, MA, USA) and observed simultaneously by fluorescence inverted microscopy [25].

#### 2.3.4. Western Blotting

The cells (1.2 × 10^6^ cells/well) were incubated into each well of a 6-well plate. The cultured cells were placed on ice and the culture medium was removed. After cells were washed with 1 mL of pre-cooled phosphate-buffered saline (PBS) 3 times, the cell lysis solution of 200 μL was added and cells were lysed on ice for 20 min. The cells on the 6-well plate were scraped by the cell scraper and transfered to a 1.5-mL Eppendorf (EP) tube. After the EP tube was centrifuged (3500, KUBOTA, Naniwa, Osaka, Japan) at 14,000 rpm at 4 °C for 10 min, the supernatant was placed into the new EP tube. The protein content of the lysate was determined using a protein assay kit (A046-1-1, Nanjing Jiancheng Bioengineering Institute, Nanjing, Jiangsu, China) following the manufacturer’s protocols. The protein loading buffer was added into the EP tube and denatured in a thermostatic metal bath at 100 °C for 7 min. After cooling to room temperature, the EP tube was stored at −20 °C until used. Proteins (20 μg/lane) were separeted on 10% sodium dodecyl sulfate (SDS) polyacrylamide gels, transferred to polyvinylidene difluoride (PVDF) membranes, and blocked with 5% skim milk powder in phosphate buffered saline with Tween-20 (PBST). The membranes were treated with a primary monoclonal antibody overnight at 4 °C before being incubated for 1 h at room temperature with a secondary antibody that had been diluted in skim milk by 2000-fold. Blots were further treated with an enhanced chemiluminescence kit (1705040, Bio-Rad, Hercules, CA, USA) and detected on an ImageQuant LAS 4000 mini (Las 4000, GE, Boston, MA, USA). A quantitative density analysis was performed using ImageJ software.

### 2.4. Transcriptome Analysis

Cells were pre-cultured on 6-well plates for 24 h. LPS (40 ng/mL) was added for 30 min, followed by 160 µg/mL and 320 µg/mL of ANP, and incubated for more than 12 h. Four groups were set up: control group (Con), LPS group (LPS), LPS + 160 µg/mL group (160/LPS), and LPS + 320 µg/mL group (320/LPS). After incubation, the culture solution was removed and washed with PBS. After the removal of PBS, RIPA lysis solution was added for 20 min. After lysis, cells were scraped off with a cell scraper. Then, the cells were collected in 1.5-mL enzyme-free centrifuge tubes. Collected cells were frozen at −80 °C. RNA extraction and detection, library preparation for transcriptome sequencing, clustering, and sequencing followed the procedures of Novogene (Beijing, China). Differential gene analysis was performed using DESeq2, for significantly different expressions, a corrected *p*-value of 0.05 and an absolute foldchange of 2 were established [26]. Gene Ontology (GO) enrichment analysis and Kyoto Encyclopedia of Genes and Genomes (KEGG) pathway enrichment analysis of differentially expressed genes were implemented by the clusterProfiler R package [27,28].

### 2.5. Data Analysis

All data were analyzed using one-way ANOVA SPSS (23, IBM, Armonk, NY, USA) and the differences among treatment groups were evaluated using least significant difference (LSD) and Duncan’s multiple range tests, with a significance threshold of *p* < 0.05 or *p* < 0.01 [29].

## 3. Results

### 3.1. Antioxidant Activity of ANP in LPS-Stimulated RAW264.7 Macrophages

#### 3.1.1. Cell Viability

RAW264.7 cells were incubated with indicated doses of ANP from 6 to 15 h and the cell viability was shown in Table 1. The cell viability was increased in a time- and dose-dependent manner. However, cell toxicity on the cells were observed when the time and concentration reached 15 h and 640 µg/mL. Therefore, we chose the dose and the time of treatment as 0–320 µg/mL and 12 h for the next experiments to avoid cellular toxicity.

#### 3.1.2. The Ability of ANP to Eliminate Intracellular ROS in RAW264.7 Cells

ROS can oxidize DCFH to generate fluorescent DCF. Thus, the fluorescence intensity is proportional to the amount of intracellular ROS [30]. The inhibitory effect of ANP on LPS-stimulated ROS in RAW264.7 cells is represented as relative DCF fluorescence and shown in Figure 1. LPS significantly increased the ROS level compared to the Con group. The addition of ANP dose-dependently reduced ROS levels induced by LPS. The data demonstrated that ANP could reduce intracellular ROS levels induced by LPS.

To confirm the cellular ROS levels after the addition of DCFH, we used fluorescence microscopy to directly observe the substance. As shown in Figure 2, LPS exhibited a markedly increased fluorescent intensity compared to that of Con. The addition of ANP dose-dependently reduced the fluorescence intensity induced by LPS. These data directly revealed that ANP could reduce intracellular ROS levels induced by LPS.

#### 3.1.3. Effect of ANP on The Expression of Antioxidant Enzymes in RAW264.7 Cells

The above data indicated that ANP could reduce ROS levels induced by LPS. To clarify whether the ROS level reduction is related to the expression of antioxidant factors or enzymes, we chose Nrf2 and HO-1 as their representatives and investigated their protein levels after treatments with 0–320 µg/mL of ANP for 12 h. As shown in Figure 3, ANP significantly enhanced Nrf2 and HO-1 levels from 40 µg/mL to 320 µg/mL of ANP.

### 3.2. Transcriptome

#### 3.2.1. The Foldchange of Differentially Expressed Genes

To evaluate the molecular mechanisms of antioxidant activity by ANP, we further investigated the effects of APN on genome-wide gene expression in RAW264 cells by high-throughput RNA-sequencing technology. The differentially expressed genes (DEGs) between each group were compared to the LPS group (Table 2 and Figure 4). The 160/LPS group generated a total of 10,504 DEGs, of which the expressions of 5263 genes were higher and the expressions of 5241 genes were lower. The log2 foldchange of three genes was higher than 10, and the variations of two hundred and eight genes were greater than 5 and less than 10. The log2 foldchange of one gene was less than −10, whereas ninety-seven genes had changes greater than −10 and less than −5. In the 320/LPS group, there were 9550 DEGs, of which 4846 genes’ expression levels increased and 4704 genes’ expression levels decreased. There were log2 foldchange of 289 genes greater than 5 and less than 10. Only one gene had a log2 foldchange less than −10, and genes had changes greater than −10 and less than −5.

#### 3.2.2. Gene Ontology Analysis

Gene Ontology (GO) is a comprehensive database that describes gene functions in three categories: biological process, cellular component, and molecular function [31]. The functions of DEGs in RAW264.7 cells following LPS stimulation were classified using GO analysis. Figure 5 depicts the results. Scatter plots were created using the 10 most prominent GO phrases for each component. Treatment with the 160/LPS were mostly targeted in the mitochondrial organization, kinase activity, and transferase activity.

While treatment with the 320/LPS highly targeted in the endoplasmic reticulum, mitochondria, protein binding, kinase activity, and transferase activity.

#### 3.2.3. Kyoto Encyclopedia of Genes and Genomes Analysis

Further pathway enrichment analysis using the Kyoto Encyclopedia of Genes and Genomes (KEGG) was performed to further understand the processes at a higher level. Table 3 depicts the adjusted *p*-value (padj) of pathways that were related to disease, inflammation, cancer, and virus infection. In the highly enriched pathways of 160/LPS, 13 pathways were enriched. In the substantially enriched pathways of 320/LPS, 16 pathways were enriched. According to the DEGs results, the pathways of 160/LPS had 9 pathways related to the MAPK pathway. In the 320/LPS results, there were 12 pathways associated with the MAPK pathway. Table 4 displayed the MAPK-related genes.

#### 3.2.4. The Interaction Networks of MAPK Pathway-Related DEGs

The protein-protein interaction network (PPI) results were based on DEGs. The protein network connections analysis was performed using DEGs from RNA-seq in 160/LPS and 320/LPS. After the original networks were built by Cytoscape, which allows visualization of molecular interaction networks, the important subnetworks and connective degrees were analyzed by MCODE and Cytohubba, which were two analysis programs in Cytoscape. Figure 6 depicts the subnetwork results of 160/LPS. Among them, 21 genes were related to the MAPK pathway and 17 genes were related to the cellular senescence pathway. A total of six genes were presented in both pathways. Table 5 shows the results of the connection degree, which indicates the degree to which the protein is connected to other proteins in the pathway.

Figure 7 depicted the subnetwork results in 320/LPS. Of these, 25 genes were associated with the MAPK pathway; 21 genes were associated with the cellular senescence pathway; and 44 genes were associated with the pathway in cancer. And seven genes were presented in three pathways. Table 6 displayed the degree of connection.

### 3.3. Inhibitory Effect of ANP on MAPK Pathway in LPS-Stimulated RAW264.7 Cells

Phosphorylation of MAPKs was reported to be strongly associated with the activity of Nrf2 [32]. The results from KEGG and PPI suggested that ANP might inhibit the MAPK signaling pathway. Therefore, we treated RAW264.7 cells with 0–320 µg/mL of ANP for 12 h, and the phosphorylation of MAPKs was then examined using Western blot to demonstrate the inhibitory impact of ANP on the MAPK signaling pathway. As shown in Figure 8, ANP inhibited the phosphorylation of MEK1, ERK, MKK3, p38, MKK4, and JNK induced by LPS in a dose-dependent manner and showed significant inhibition at 320 µg/mL of ANP.

## 4. Discussion

Areca nuts are a popular tropical fruit in Asian areas, and they are rich in polyphenols. However, there are limited investigations on the benefit to health because areca nut contains some carcinogenic alkaloids, such as arecoline, arecaidine, and guvacine. The areca nut alkaloids have been reported to have reproductive toxicity and ROS production [33,34]. On the other hand, the extracts of areca nut were reported to have antioxidant potential [35,36], and antibacterial activity [37,38].

In our previous study, two kinds of polyphenols, catechin and proanthocyanidin B1 were found [23]. To further investigate the antioxidant activity of ANP, we used a macrophage-like cell line, RAW264.7, which is a cellular model to study antioxidant and anti-inflammatory activities at the cellular level. As a result, ANP had a significant inhibitory effect on LPS-stimulated ROS (Figure 1 and Figure 2). Moreover, ANP significantly increased Nrf2 and HO-1 levels (Figure 3). Moreover, continuous stimulation of LPS causes inflammation of the cells, and MAPK pathway is associated with the production of inflammation [39,40,41]. Our data revealed that ANP further downregulated the activation of the MAPK signaling pathway (Figure 8). Thus, ANP inhibited not only LPS-stimulated oxidative stress, but also the LPS-stimulated inflammation.

In order to further investigate how APN affects gene expression across the entire genome in RAW264 cells; we used RNA sequencing and bioinformatics. Our data revealed that ANP had a substantial influence on mitochondria, kinase activity, and transferase activity according to GO analysis (Figure 5). Therefore, we speculate that ANP may alleviate oxidative stress by regulating mitochondrial activity and protease activity. KEGG analysis revealed that ANP affected some pathways that are related with diseases, NAFLD, virus infections, and bacterial infections. Most of them are related to the MAPK pathway (Table 3). ANP inhibited the expressions of MAPK-related genes (Table 4). These data also supported the finding that ANP prevented LPS-stimulated oxidative damage via the MAPK signaling pathways. A similar pathway was also reported: cryptochlorogenic acid activated the Nrf2/HO-1 signaling pathway via the ROS-dependent MAPK pathway, and reduced the LPS-stimulated inflammation [42].

In the PPI analysis according to the DEGs data, we found that the cellular senescence pathway and the MAPK pathway interacted with each other (Figure 6). In the 160/LPS, Mapk3, which is also known as Erk1 and activates the activity of many MAPK enzymes, had the highest connectivity degree and was associated with most of the genes. Erk1/2 is reported to play a key role in cellular senescence [43]. In the 320/LPS, Mapk1, which is also known as Erk2, had the highest connectivity degree, can activate the activity of many protein kinases, and is involved in a variety of cellular processes (Figure 7). A previous study showed that Erk1/2 and p38 play a key role in cellular senescence due to smoking [44].

## 5. Conclusions

In conclusion, our work demonstrated that ANP reduced LPS-stimulated oxidative stress through the MAPK-mediated Nrf2/HO-1 signaling pathway. These data provide the molecular bases for understanding the antioxidant activity of ANP.

## Figures and Tables

**Figure 1 foods-11-03607-f001:**
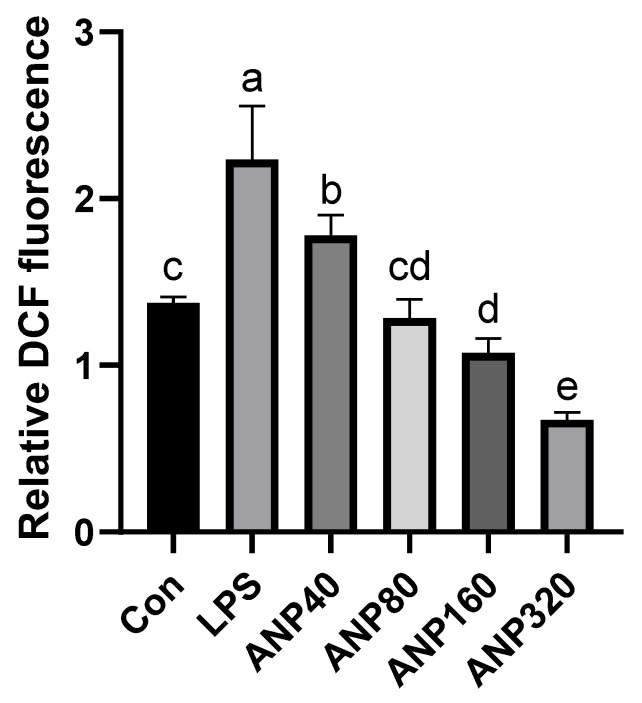
Inhibitory effect of ANP on the intracellular ROS generation. The results shown are expressed as means ± SD (*n* = 3). The different letters in the graph indicated significance (*p* < 0.05).

**Figure 2 foods-11-03607-f002:**
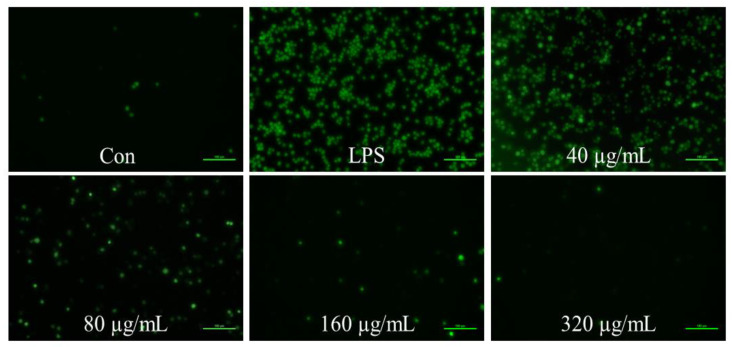
Observation of cellular ROS by fluorescence inverted microscope.

**Figure 3 foods-11-03607-f003:**
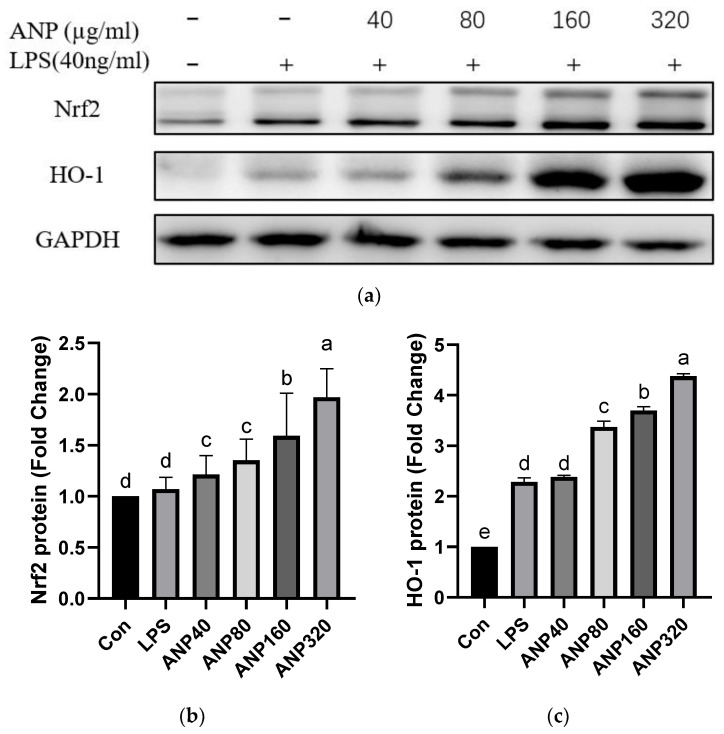
ANP enhancement of the Nrf2 and HO-1 proteins: (**a**) Nrf2 and HO-1 were detected by Western blot and GAPDH was used as a control; (**b**,**c**) the densitometric quantification of bands showing Nrf2 and HO-1. The results shown are presented as means ± SD (*n* = 3). The different letters in the graph indicate significance (*p* < 0.05).

**Figure 4 foods-11-03607-f004:**
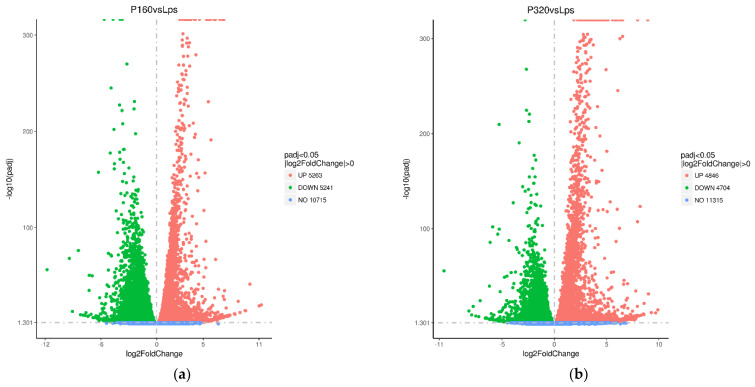
The differentially expressed genes volcano plots of 160/LPS (**a**) and 320/LPS (**b**). The padj is a *p*-value after correction for multiple hypothesis testing.

**Figure 5 foods-11-03607-f005:**
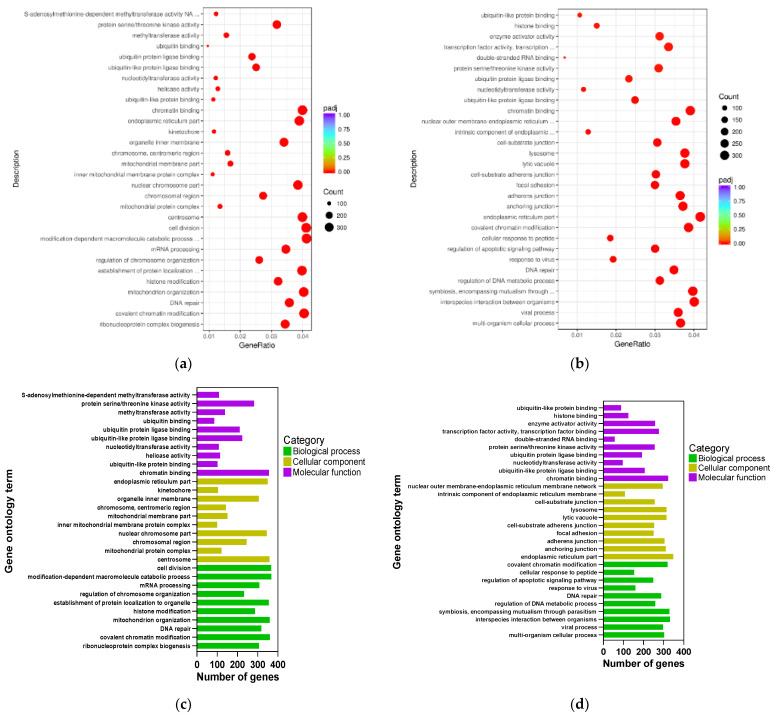
The GO analysis of all DEGs (**a**) gene ratio in 160/LPS, (**b**) gene ratio in 320/LPS, (**c**) number of genes in 160/LPS, (**d**) number of genes in 320/LPS. The padj is a *p*-value after correction for multiple hypothesis testing. The gene ratio is a ratio of the number of differential genes annotated to the GO term to the total number of differential genes.

**Figure 6 foods-11-03607-f006:**
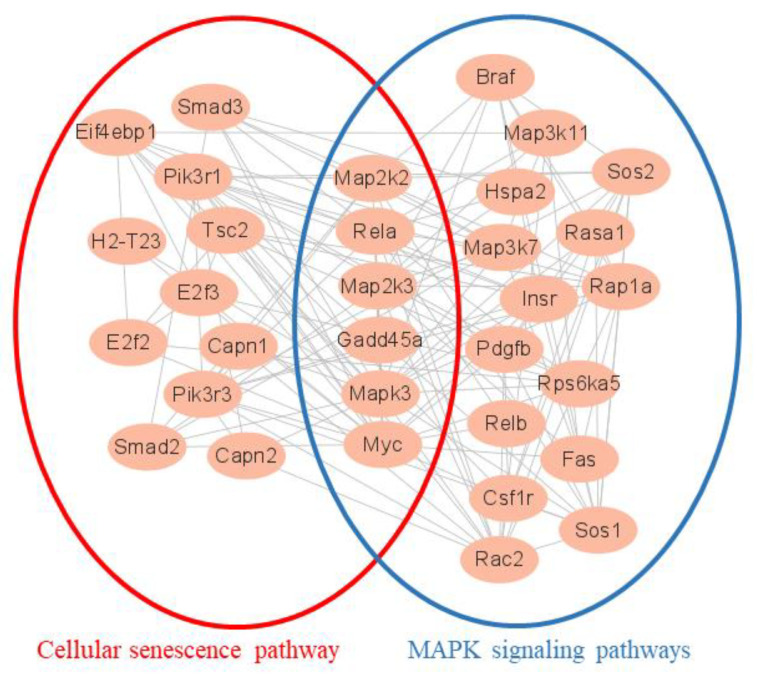
The protein-protein interaction subnetwork of all DEGs network in 160/LPS.

**Figure 7 foods-11-03607-f007:**
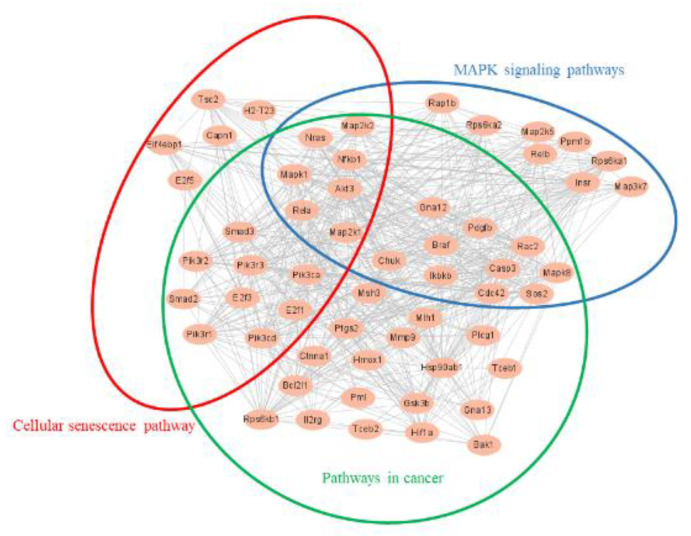
The protein-protein interaction subnetwork of all DEGs network in 320/LPS.

**Figure 8 foods-11-03607-f008:**
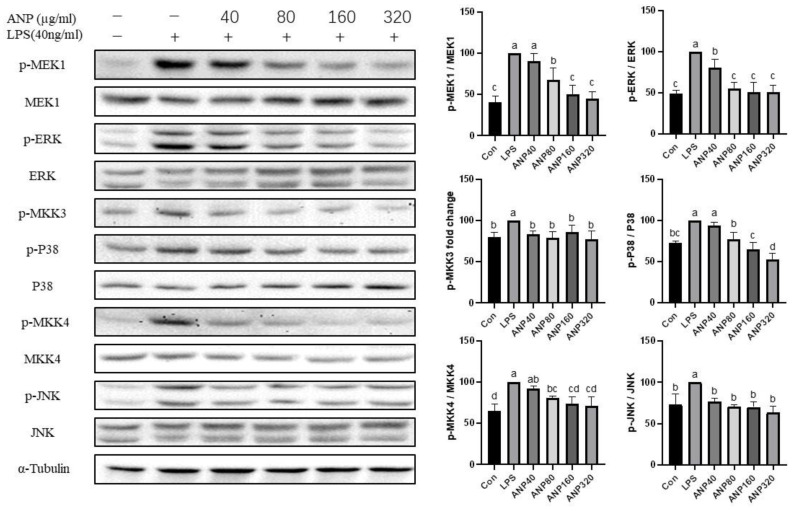
ANP inhibited the phosphorylation of MEK1, ERK, MKK3, p38, MKK4, and JNK induced by LPS. **Left** pictures are western blots of indicated proteins and α-tubulin as a control. **Right** pictures are the density histogram of western blots. The “*p*-” means “phosphorylated”. The results shown are presented as means ± SD (*n* = 3). The different letters in the graph indicate significance (*p* < 0.05).

**Table 1 foods-11-03607-t001:** The effect of areca nut polyphenol extract on the viability of RAW264.7 cell.

Item	Concentration	6 h	9 h	12 h	15 h
Control	-	1.00	1.10 ± 0.03	1.16 ± 0.04	1.23 ± 0.06
ANP	40 µg/mL	1.02 ± 0.01	1.12 ± 0.10	1.20 ± 0.09	1.17 ± 0.05
ANP	80 µg/mL	1.07 ± 0.05	1.15 ± 0.11	1.25 ± 0.06	1.21 ± 0.06
ANP	160 µg/mL	1.04 ± 0.06	1.18 ± 0.05	1.27 ± 0.07	1.15 ± 0.05
ANP	320 µg/mL	1.02 ± 0.06	1.20 ± 0.01	1.20 ± 0.01	1.11 ± 0.07 ^#^
ANP	640 µg/mL	1.01 ± 0.01	1.06 ± 0.10	1.03 ± 0.08 *	1.09 ± 0.04 ^#^

Five replicates were used in the experiment. According to ANOVA, the means with different letters showed significant differences (*p* < 0.05), * compared with the 12 h control group, # compared with the 15 h control group. “-” means the concentration of ANP is zero.

**Table 2 foods-11-03607-t002:** The statistical table of differentially expressed genes.

Log2 FoldChange	160/LPS	320/LPS
≥10	3	0
≤5 to <10	205	289
≤2 to <5	1061	1295
<−2 to <2	7655	7035
<−5 to ≤−2	1483	876
<−10 to ≤−5	96	54
≤−10	1	1

**Table 3 foods-11-03607-t003:** The significantly enriched KEGG pathways in the treatment of 160/LPS and 320/LPS.

Pathway	160/LPS	320/LPS
NOD-like receptor signaling pathway	0.000182	0.0001367
Epstein-Barr virus infection	0.000229	0.0075382
Parkinson’s disease	0.005891	0.0016235
Alzheimer’s disease	0.015089	0.0003044
Non-alcoholic fatty liver disease (NAFLD)	0.015089	0.0000129
Herpes simplex infection	0.015089	0.0068945
Huntington’s disease	0.015089	0.0150887
Salmonella infection	0.017324	-
Tuberculosis	0.020583	0.0137757
Endometrial cancer	0.036453	-
Cellular senescence	0.036453	-
TNF signaling pathway	0.043539	0.006894
Kaposi’s sarcoma-associated herpesvirus infection	0.047895	0.006997
Influenza A	-	0.003123
Chagas disease (American trypanosomiasis)	-	0.003123
Hepatitis B	-	0.004054
Hepatitis C	-	0.032155
Pancreatic cancer	-	0.047466
Toll-like receptor signaling pathway	-	4.15E-05

The “-” sign means the pathway was not significantly enriched.

**Table 4 foods-11-03607-t004:** The MAPK pathway-related DEGs in the treatment of 160/LPS and 320/LPS.

Gene Name	Fold Change	
	160/LPS	320/LPS
**p38-related gene**		
Mitogen-Activated Protein Kinase 14 (Mapk14)	−1.13	−1.68
Mitogen-Activated Protein Kinase 11 (Mapk11)	−1.43	−1.54
**JNK-related gene**		
Mitogen-Activated Protein Kinase 9 (Mapk9)	−2.35	−2.73
Mitogen-Activated Protein Kinase 8 (Mapk8)	−1.84	−1.10

The “−” sign means the fold change was downregulated.

**Table 5 foods-11-03607-t005:** The six hub genes with the greatest degree of connection in subnetwork of 160/LPS.

Gene Symbol	Gene Description	Degree
Mapk3	Mitogen-activated protein kinase 3	20
Myc	MYC proto-oncogene	18
Rela	RELA proto-oncogene	12
Map2k2	Mitogen-activated protein kinase kinase 2	11
Map2k3	Mitogen-activated protein kinase kinase 3	9
Gadd45a	Growth arrest and DNA-damage-inducible 45 alpha	2

**Table 6 foods-11-03607-t006:** The seven hub genes with higher degrees of connectivity.

Gene Symbol	Gene Description	Degree
Mapk1	Mitogen-activated protein kinase 1	33
Akt3	AKT serine/threonine kinase 3	32
Nfkb1	Nuclear factor kappa B subunit 1	32
Rela	RELA proto-oncogene	25
Nras	NRAS proto-oncogene	24
Map2k1	Mitogen-activated protein kinase kinase 1	22
Map2k2	Mitogen-activated protein kinase kinase 2	22

## Data Availability

All relevant data are presented in this study.

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
