# Peer review of "In Vitro Antioxidant Activity of Areca Nut Polyphenol Extracts on RAW264.7 Cells"

_foods, 2022, doi:10.3390/foods11223607_

Round 1

Reviewer 1 Report

I reviewed the manuscript entitled, In vitro antioxidant activity of areca nut polyphenol extracts on RAW264.7 cells. The manuscript has no novelty and no scientific soundness. Authors simply conducted the antioxidant analysis of nut polyphenols. There are many studies in the literature dealing with antioxidant activity of nut polyphenols. Authors used RAW264.7 cells. For example, studies from 2000-2015 investigated In vitro antioxidant activity of areca nut polyphenol extracts (https://doi.org/10.3390/molecules191016416; https://www.ajol.info/index.php/ajb/article/view/62076 ; https://doi.org/10.5530/ax.2011.1.6 )

Authors failed to perform HPLC analysis to determine polyphenols and no purification studies. And, used very basic antioxidant analysis.

Reviewer 2 Report

The study aims to investigate the antioxidant activity and its mechanisms of areca nut polyphenol fraction in LPS-activated RAW264.7 cells by applying ROS measurement, gene and protein expression analysis. It supplements the study field with valuable information, the study is novel; however, the presentation of the results is of low quality and numerous corrections of manuscript must be made. Extensive editing of English language and style is required.

Abstract

First sentence must be revised (“popular used”, “..nut is …a hobby…”, “…rich in polyphenol…” (which one?)).

The abbreviation LPS is not given in fa ull form.

Introduction

Nrf2, c-Jun, ATF2, and ATF4 should be given in a full form.

The abbreviation LPS is not given in a full form and the reason of activation of RAW264.7 cells with LPS is not explained. Also, another term for LPS-activated RAW264.7 cells is used further in the manuscript (“LPS-induced RAW264.7”) which should be changed.

Materials and Methods

Section 2.1. The type and manufacturer of secondary antibodies is missing.

Section 2.2. The concentration of ethanol, extraction duration and resin type are missing. The use of “again” in line 86 is unclear. The statement “80% areca nut polyphenols were obtained” is unclear – is 80% the recovery or the purity of the crude extract and how it was evaluated?

The title of Section 2.3. is “Antioxidant activity assays in LPS-induced RAW264.7 macrophages”, however, the procedure of induction or activation by LPS is missing.

Section 2.3.2. must be rewritten due to bad English (“…assay was measured by … assay”, etc.) and incorrectly described procedures. In the last sentence, there is no centrifugation procedure described, however supernatant was removed; formazan solution was added what is illogical since formazan is already formed in the cells and should be measured. What was detected at 490 nm?

Section 2.3.3. It is unclear whether the cell fluorescence was measured in dry cells. “Fluorescent counter” should be changed to “Fluorescence counter”.

Section 2.3.4. The abbreviations PBS, EP, and PBST are not given in a full form. Centrifugation units must be g or, if it is rpm, then rotor diameter or the model of a centrifuge should be provided. It is unclear, what conditions should be kept between two denaturation steps (line 128). The proteins investigated by Western blot are not indicated.

Section 2.4. The concentration of LPS and treatment duration is missing. Cell collection method is missing. Sentence in lines 146-147 should be more detailed. Last sentence is incomplete. The information on threshold of fold change in upregulation and downregulation is missing.

Results

All the titles of graphs and tables are either incorrect or incomplete!

Break-up into Sections 3.1.2.1-3.1.2.1. is redundant.

In Figure 3, graphs aren’t numbered. GAPDH is measured, however, the result is not mentioned neither in figure title, nor in Results or Discussion section.

Section 3.2.1. Section title is incorrect. The terms used in the section are inappropriate. What authors mean by gene ploidy? The descriptions „elevated“, „decreased“, „raised“, „lowered“, „created“ are not accurate for genes (DEGs). The full form of DEGs „differential expression genes“ is incorrect.

The text in figures 4 and 5 is too small therefore unreadable in printed version.

Incomplete sentences in lines 253-254, 265, 269.

Section 3.3., line 282, “…at least 320” – the concentration should be checked.

Discussion

Line 299, the statement “we separated the polyphenol part of areca nut without or few alkaloids” is not supported neither by the results of this manuscript nor by any reference.

Conclusions

The statement “…ANP reduced LPS-induced oxidative damage” should be revised since there were no measurements of “damage”. Increased ROS generation do not automatically mean damage to cell.

Reviewer 3 Report

This manuscript describes the antioxidant activity of areca nut polyphenol extracts on

RAW264.7 cells. Some comments needed to be considered:

1. Abbreviations should be mentioned in the abstract as full names at the first time. Also, all abbreviations in the manuscript should be described after the keywords.

2. English should be checked by a native speaker. line 17, change Asia to Asian,

line 44 omit (lines of), line 86 omit (again).

3. line 83, the name and affiliation of the taxonomist who identified and authenticated the plant should be described.

4. Voucher specimen code and season of collection of plant material should be stated.

5. line 88, the methodology for obtaining polyphenol rich extract through macroporous resin is not clear. Kindly desribe and clarify

6. Line88, how did the authors quantify the polyphenols content as 80%?

7. lines 171,172: Kindly add a citation to them

8.Figures 1 and 2, authors should add ANP on the figures beside the doses used 

9.Figure 3, letters on the figure are not described in the figure caption and some SD values are relatively high 

10. line 208, change Tables to Table. Also, figure 4 is not cited in the text of the manuscript.

11. line 269, change (degree) to (degree of connectivity) to better describe the table

Round 2

Reviewer 1 Report

I re-reviewed the manuscript entitled, In vitro antioxidant activity of areca nut polyphenol extracts on RAW264.7 cells. Authors mentioned the polyphenol purity at 80 %. How Folin-Ciocalteu method can determine the purity of polyphenols. It is used to determine the TPC of samples.

Authors also mentioned in their response that they already used HPLC analysis. I suggest authors include ref and provide what are the major phenolic compounds. Then authors can say polyphenol extracts  

Reviewer 2 Report

Comments are in a pdf file attached. Minor revision is needed.
